# Intermittent Stop-Move Motion Planning for Dual-Arm Tomato Harvesting Robot in Greenhouse Based on Deep Reinforcement Learning

**DOI:** 10.3390/biomimetics9020105

**Published:** 2024-02-10

**Authors:** Yajun Li, Qingchun Feng, Yifan Zhang, Chuanlang Peng, Chunjiang Zhao

**Affiliations:** 1College of Mechanical and Electrical Engineering, Hunan Agriculture University, Changsha 410128, China; lyj20210043@stu.hunau.edu.cn; 2Intelligent Equipment Research Center, Beijing Academy of Agriculture and Forestry Sciences, Beijing 100097, China; vishkcc@163.com (Y.Z.); pcl109969@163.com (C.P.); 3Beijing Key Laboratory of Intelligent Equipment Technology for Agriculture, Beijing 100097, China

**Keywords:** motion planning, task allocation, deep reinforcement learning, dual-arm harvesting robot

## Abstract

Intermittent stop–move motion planning is essential for optimizing the efficiency of harvesting robots in greenhouse settings. Addressing issues like frequent stops, missed targets, and uneven task allocation, this study introduced a novel intermittent motion planning model using deep reinforcement learning for a dual-arm harvesting robot vehicle. Initially, the model gathered real-time coordinate data of target fruits on both sides of the robot, and projected these coordinates onto a two-dimensional map. Subsequently, the DDPG (Deep Deterministic Policy Gradient) algorithm was employed to generate parking node sequences for the robotic vehicle. A dynamic simulation environment, designed to mimic industrial greenhouse conditions, was developed to enhance the DDPG to generalize to real-world scenarios. Simulation results have indicated that the convergence performance of the DDPG model was improved by 19.82% and 33.66% compared to the SAC and TD3 models, respectively. In tomato greenhouse experiments, the model reduced vehicle parking frequency by 46.5% and 36.1% and decreased arm idleness by 42.9% and 33.9%, compared to grid-based and area division algorithms, without missing any targets. The average time required to generate planned paths was 6.9 ms. These findings demonstrate that the parking planning method proposed in this paper can effectively improve the overall harvesting efficiency and allocate tasks for a dual-arm harvesting robot in a more rational manner.

## 1. Introduction

The proliferation and advancement of industrial greenhouse cultivation models have significantly boosted the mass and continuous production of fruits and vegetables, with cherry tomatoes yielding 48–57 kg/m^2^ [1,2]. In such high-yield environments, the robotic farming equipment has become essential for complex and labor-intensive horticultural tasks, enhancing efficiency and optimizing production processes [3]. Commercial fruit-harvesting robots have been developed and deployed using common fresh fruits such as tomatoes [4], strawberries [5], and cucumbers [6] as targets.

In past years, a growing number of researchers have incorporated multiple operational units, and adopted multi-arm collaborative task-planning methodologies, to enhance the operational efficiency of robots [5,7]. This integration has positioned multi-arm robotic systems as a promising technological solution, with some already commercially available [8]. While these multi-arm robots have offered benefits in workspace utilization and operational efficiency, they have also faced more complex, higher-dimensional technical challenges. In traditional robotic harvesting processes, robots stop and perform harvesting at each encountered target. The parking decision method did not take into account the regional growth distribution characteristics of fruits on either one or both sides. Instead, they focused solely on whether there was a harvestable fruit present to make the decision. Dual-arm robots required more efficient parking nodes planning and collaborative strategies compared to traditional single-arm robots and the aforementioned stopping methods [9]. There were two primary reasons for this. Firstly, different targets could result in significant variations in operation time. For instance, if the distribution of fruits was uneven, one or several arms might remain idle until the others complete their tasks. Secondly, the robotic platform experienced time wastage while switching the working areas. A harvesting robot, moving along crop rows and determining the appropriate parking node for operation, must consider both the overall operational time and the number of necessary stops. Therefore, less frequent parking node planning on the part of the multi-arm robot vehicle is crucial for harvesting collaboration, aiming to maximize overall efficiency and avoid idle periods in arm operation.

Parking node planning for multi-arm robots can be viewed as a decision-making problem, akin to the multiple Traveling Salesman Problem (mTSP) or Vehicle Routing Problem (VRP). This issue has necessitated description and resolution through applications of operations research and mathematical programming techniques [9]. Barnett et al. [7] devised a task distribution strategy for multi-arm kiwifruit harvesting robots, considering both task partitioning and reachability. This strategy aims to ensure uniformity in fruit distribution while minimizing the time required to complete the task. Wang et al. [10] introduced an improved ant colony algorithm that focuses on the entire work path, facilitating whole-process path planning for multiple agricultural machines. This approach resulted in a reduction in path cost by 14% to 33%. Furthermore, some scholars have developed parking node planning algorithms using techniques such as grid maps [11], spiral-spanning tree coverage (STC) [12], area division [13], etc. These algorithms have the potential to significantly improve the working efficiency of harvesting robots [9]. However, vehicle parking node planning for multi-arm robots harvesting in multiple directions have presented challenges due to additional constraints such as task allocation, workspace, stop frequency, etc.

Deep Reinforcement Learning (DRL), a type of machine learning algorithm where agents learn optimal behaviors through trial-and-error interactions with their environment, have demonstrated exceptional performance in robotics [9,14], path planning [2,15], and combinatorial optimization problems [16,17]. Kyaw et al. [18] employed DRL based on grid maps to solve the Traveling Salesman Problem (TSP), achieving path planning for robots in large, complex environments. Considering the capability of DRL to address optimization decision-making problems with less prior information, Martini et al. [19] implemented position-agnostic autonomous navigation in vineyards using DRL, without the help of GPS and visual odometry technologies. Wang et al. [13] developed a DRL-based movement planning method for kiwifruit-harvesting robots, converting traditional grid-based coverage path planning into a TSP problem focused on area traversal order. This method enhanced navigation efficiency by 35.72% over the boustrophedon algorithm. These DRL-based methods achieved a robust and efficient performance for various robot controllers and planners [2]. However, these methods primarily targeted the path planning of the robot vehicles with a single arm or multiple arms while harvesting on the same side [9,13]. They were not suitable for multi-arm harvesting robots operating on both sides simultaneously in a greenhouse environment.

In this paper, we introduced a Deep Reinforcement Learning-based minimal frequency parking node planning algorithm for a dual-arm harvesting robot, specifically developed for tomato greenhouses. The algorithm simultaneously optimizes two primary objectives, minimum parking frequency and fruit distribution on both arms, to maximize the overall operational efficiency of the robot. Our approach employed a “mapping-then-motion-planning” strategy. The dual-arm robot began by moving from the starting to the endpoint of the crop row to construct a projected map of tomato clusters on both sides. This projection map was then input into a policy model to generate a reverse movement path for the robot vehicle. The goal was to enable the agent to take into account the overall distribution comprehensively, minimize unnecessary parking, and address the issue of uneven fruit allocation leading to idle in the arms. The main contributions are as follows:(1)A real-time tomato cluster coordinate projection method was proposed to obtain the distribution of tomato clusters in a greenhouse environment;(2)A deep reinforcement learning algorithm, DDPG, was employed to generate the sequence of parking nodes for the vehicle of a dual-arm harvesting robot. This replaced the conventional method whereby tomato-harvesting robots would cease and harvest at each target they encountered. This novel approach ensured that both arms were actively engaged in harvesting tasks during each parking phase, with each arm executing multiple harvests;(3)The trained policy model was implemented on an actual dual-arm robot and compared with a grid-based and area division parking node planning algorithm. The feasibility of real-time mapping and efficient parking planning was demonstrated. While ensuring no tomatoes were missed, this approach addressed the main limitation of existing planning methods, which failed to consider the simultaneous operation of both arms of the robot. This significantly reduced the frequency of the vehicle’s stops, thereby enhancing the overall harvesting efficiency and enabling effective task distribution between the two arms of the robot.

## 2. Materials and Methods

### 2.1. Greenhouse Environment

The harvesting experiments were conducted in the Shounong Cuihu Agricultural Tomato Production Park in Haidian District (116.176242 E, 40.127529 N), Beijing, China, depicted in Figure 1. This greenhouse adhered to the planting and management practices typical of the Dutch standard. The layout featured dual crop row arrangements separated by aisles, facilitating robot movement along the rows. Each row comprised plants suspended from gutters attached to the roof of the greenhouse [20]. Notably, workers had previously pruned the leaves in the height range of existing mature fruits, to improve ventilation and sunlight and reduce pests and diseases [21]. Given the natural variability in plant growth and ongoing maintenance activities, we conducted measurements and analyses on 10 distinct crop rows. Consequently, the identified harvesting area was characterized by a ground clearance height ranging from 100 to 148 cm and a width of 20 cm. The spacing of crop rows measured 91.5 cm on the aisle side and 70 cm on the backside, with a length of 100 m. The practical aisle width was approximately 71.5 cm, considering the encroachment of plant fruits and leaves, which further limited the available space.

### 2.2. Dual-Arm Harvesting Robot System and Workspace Area

Figure 2 shows our dual-arm robot prototype, which consisted of an orbital-type mobile vehicle equipped with two seven-degrees-of-freedom (7-DOF) arms (xMateER3Pro, Rokea, Beijing, China). The arms were arranged in a staggered configuration on the vehicle with a longitudinal spacing of 0.9 m. The height of the arm base from the ground was 0.69 m. The end-effector was attached to, and therefore a part, of Link 7. The link coordination system of the arm is shown in Figure 2c. An Intel Realsense D435i RGB-D camera (Intel Corporation, Santa Clara, CA, US) was fixed on the end-effector of the arm in an eye-in-hand configuration, facing the plants on both of the robot’s sides.

In our study, traditional grasping methodologies [4,22] were enhanced through the implementation of an adaptive adjustment mechanism in the end-effector. This modification allowed for precise alignment with the fruit main-stem and peduncle during the harvesting process. Notably, the 6th joint of the arm consistently maintained a horizontal orientation, as illustrated in Figure 3a. Considering the above pose constraints of the arm, a Monte Carlo method was used to calculate the dexterous workspace [23] of the arm, and randomly generated 50,000 points. As shown in Figure 3b–d, the diameter of the dexterous workspace of the arm was nearly 1.0 m. The center of the harvesting area was positioned 0.7 m from the arm base, enabling it to cover the entire area effectively. Considering the height range and width of the harvesting area, the actual effective workspace area of the arm was projected onto a two-dimensional plane. This projection, shown as the red area in Figure 3e, outlined a rectangular region with dimensions of 0.439 × 0.48 m. However, this study utilized a slightly smaller area of 0.4 × 0.48 m for analysis.

### 2.3. Tomato Cluster Coordinate Projection and Mapping

This study referenced SLAM (Simultaneous Localization and Mapping)-based navigation techniques [5], implementing a “mapping-then-motion-planning” strategy. This approach was designed to enable the harvesting robot to efficiently plan its movements, taking into account the distribution of ripe tomato clusters across the two rows.

As demonstrated in Figure 4, the robot simultaneously captures color and aligned depth images of the crop rows on both sides using dual RGB-D cameras during movement. Each RGB image boasted a resolution of 640 × 480 pixels. The cameras were positioned at a height of 1.24 m and were located 0.6 m from the center of the harvesting area. The YOLOv5 deep learning network [21,24] was used to identify the tomato clusters in real time. The model was designed to identify three distinct types of tomato clusters: unripe, ripe, and ripe yet unsuitable for harvesting due to obstructions. As the cameras traveled along the crop rows at 2 m/s in the tomato greenhouse (Section 2.1), the detection model attained a precision of 94.4% and an F1-score of 95.7% (Figure 4d). Furthermore, our model demonstrated the capability for real-time detection, operating efficiently at a rate of 16.5 frames per second (FPS) [21].

In greenhouses, tomatoes were grown at a high density with minimal spacing between tomatoes on adjacent crop rows [24]. Consequently, when images were acquired from the current crop row, tomato clusters from the opposite row may have appeared in the field of view. To address this, our approach involved capturing the depth information (Z) of the ripe tomato clusters in the images. We then filtered and retained only those ripe clusters that fell within a specific depth threshold (0.4 m < Z < 0.8 m) and were unobstructed (Figure 4e).

To reduce repeated localization of the same tomato cluster, our approach incorporated the use of ByteTrack [24,25] for tracking the tomato clusters that were preserved following depth threshold processing. The ByteTrack method was straightforward yet effective, utilizing only the Kalman filter to predict the trajectory of objects in the current frame for the next frame. It then calculated the Intersection over Union (IoU) between the predicted boxes and the detected boxes as a measure of similarity for matching.

The ByteTrack method could identify the locations and IDs of tracked tomato clusters in the current frame. However, the tomato clusters shift from the edges to the center of the image as the robot moves, causing significant changes in the bounding box proportions [24]. This shift could lead to the displacement of the same tomato cluster’s ID across consecutive frames. With reference to [7], we designed a tomato clusters position projection and mapping method based on a specific tracking area. The detailed steps can be described as follows:

First, we designated a specific area in the image (the red transparent rectangular area formed by $\left(x_{l},y_{l}\right)$ and $\left(x_{r},y_{r}\right)$ in Figure 4f), reducing the tracking area to minimize changes in the bounding box ratio. Specifically, we divided the area into left and right sections based on the center line $x_{c}=\left(x_{r}+x_{l}\right)/2$;Then, the outputs by ByteTrack within the specific area were divided into two types—the first type included tomato clusters with centroids $(x_{i}^{c},y_{i}^{c})\;$on the left side of the specific area, and another type was tomato clusters on the right side;Next, we continuously updated the IDs of tomato clusters in the left and the right of the specific area in real time. If a new tomato cluster ID moved from the left section to the right section or vice versa (Algorithm 1), we outputted the centroid coordinates P of the tomato cluster, along with the current position of the vehicle through an Xsens MTi-30 IMU (Inertial measurement unit).

**Algorithm 1:** The tomato cluster tracking method.**Input**: IDi; (xic,yic); (xl,yl); (xr,yr); xc; *Array_left_*; *Array_right_*; P**Output**: *Array_left_*; *Array_right_;*P1:**if** xl≤xic≤xc and yl≤yic≤yc **then**2:**if** IDi not in *Array_left_* **then**3:Add IDi into *Array_left_*4:**if** IDi in *Array_right_* **then**5:P←(xic,yic)6:**end if**7:**end if**8:**else if** xc≤xic≤xr and yl≤yic≤yr **then**9:**if** IDi not in *Array_right_* **then**10:Add IDi into *Array_right_*11:**if** IDi in *Array_left_* **then**12:P←(xic,yic)13:**end if**14:**end if**15:**end if**16:**return** *Array_left_*; *Array_right_;*P

To project the tomato clusters in a two-dimensional map, hand-eye calibration [26,27] was applied to transform the output centroid coordinates P of the fruit from the Camera Coordinate System to the Robot Base Coordinate System. At time *t*, the coordinates *M^L^*, *M^R^* of the fruits in the crop rows on both sides, within the map coordinate system, were represented as shown in Equation (1). By controlling the movement of the mobile vehicle to traverse the crop rows, we achieved fruit projection effects as shown in Figure 4f. Here, the black points represent the projected coordinates of tomato clusters on the two-dimensional map.
(1)$$\left\{\begin{array}{l} M^{L}=\left(D_{t}-P_{Y}^{B1},P_{Z}^{B1}\right)\\ M^{R}=\left(D_{t}+d_{0}+P_{Y}^{B2},{-P}_{Z}^{B2}\right) \end{array}\right.,$$
where $\left({P_{X}^{B1},P}_{Y}^{B1},P_{Z}^{B1}\right)$ denotes the position of the left-side target tomato cluster relative to the robot base coordinate system of the B1 arm. $\left({P_{X}^{B2},P}_{Y}^{B2},P_{Z}^{B2}\right)$ denotes the position of the right-side cluster relative to the B2 arm. $D_{t}$ signifies the distance the vehicle has moved at time *t*, and $d_{0}$ is the between the B1 and B2 arms. The B1 and B2 arms are shown in Figure 2.

### 2.4. Infrequency Movement Planning for Multi-Arm Robot Vehicle

Our objective was to transform traditional grid-based vehicle movement planning into an RL process. After the robot acquired the projected coordinates of fruit clusters, we planned a minimal frequency movement path for a dual-arm vehicle in the two-dimensional map. The m movement path typically consisted of a sequence of parking nodes. Any node on a path was always dependent on the environment and its previous node. Consequently, vehicle movement planning was achieved via a Markov Decision Process (MDP). The DRL algorithm, which iteratively improved the decision performance through explorations and thus adapted to different environments, showed certain advantages in solving MDP problems. This section first introduced a cutting-edge DRL algorithm: the Deep Deterministic Policy Gradient algorithm (DDPG). A learning strategy for training the DDPG within a specialized interactive environment was then discussed, followed by the relevant observed state and the reward function.

#### 2.4.1. Background of DDPG

The DDPG algorithm, as a model-free, off-policy, policy-based method in DRL, has garnered recognition for its effective integration of Q-learning with deterministic strategy optimization. Leveraging deep neural networks for function approximation and employing strategies such as experience replay and target networks, DDPG enhanced both stability and efficiency in learning [28]. This approach has demonstrated significant efficacy in handling continuous control challenges within robotic systems [15,29].

The DDPG algorithm belongs to a family of actor-critic reinforcement learning algorithms that aim to maximize the expected cumulative long-term reward. There are four neural networks in the algorithm: (1) The critic network (Q network), $Q\left(s,a|\theta^{Q}\right)$; (2) the target critic network $\bar{Q}\left(s,a|\theta^{\bar{Q}}\right)$; (3) the actor network (deterministic policy network) $\mu \left(s|\theta^{\mu }\right)$; (4) the target actor network $\bar{\mu }\left(s|\theta^{\bar{\mu }}\right)$. Neural networks were here characterized by parameters denoted as *θ*, with superscripts used to identify the specific network.

The critic network $Q\left(s,a|\theta^{Q}\right)$ was updated by minimizing the loss function, which was often a mean squared error between the main Q function and the target Q-values,
(2)$$L\left(\theta^{Q}\right)=E_{\left(s_{t},\;a_{t},r_{t},\;s_{t+1}\right)\sim D}\left[{\left(Q\left(s_{t},a_{t}|\theta^{Q}\right)-y_{t}\left(r_{t},s_{t+1}\right)\right)}^{2}\right],$$
with the target Q-values $y_{t}$ constructed based on the Bellman equation:(3)$$y_{t}\left(r_{t},s_{t+1}\right)=r_{t}+\gamma \bar{Q}\left(s_{t+1},\bar{\mu }\left(s_{t+1}|\theta^{\bar{\mu }}\right)|\theta^{\bar{Q}}\right),$$
where D denotes the replay buffer to store experience tuples $\left(s_{t},a_{t},r_{t}s_{t+1}\right)$.$r_{t}=r\left(s_{t},a_{t}\right)$ denotes the reward that was obtained when the agent executed action $a_{t}$ from the action space A while in state $s_{t}$, belonging to the state space S. γ is a discount factor in the range of 0 to 1, but usually close to 1 [30].

The actor network $\mu \left(s|\theta^{\mu }\right)$ was updated using the deterministic policy gradient:(4)$$\nabla_{\theta^{\mu }}J=E_{s_{t}\sim D}\left[{\left.\nabla_{a}Q\left(s,a|\theta^{Q}\right)\right|}_{s=s_{t},a=\mu \left(s_{t}\right)}{\left.\nabla_{\theta^{\mu }}\mu \left(s|\theta^{\mu }\right)\right|}_{s=s_{t}}\right],$$
To stabilize the learning process, the weights of the target networks $\bar{Q}$ and $\bar{\mu }$ were softly updated to slowly track the values of $\theta^{Q}$ and $\theta^{\mu }$. Detailed information about this procedure can be found in [28].

In particular, to ensure exploration in the DDPG framework with a deterministic policy, noise was added to each action. In this study, we used uncorrelated, mean-zero Gaussian noise to optimize the learning process [30]. The final action at time *t* was computed using the formula that incorporates the noise $N_{t}$:(5)$$a_{t}=\mu \left(s|\theta^{\mu }\right)+N_{t}.$$

#### 2.4.2. Training DDPG for Movement Planning

##### The Interaction Environment

To solve the movement planning problem within the DRL framework, we built a two-dimensional simulation environment to collect a sufficient number of samples to train DDPG. The positions of tomato clusters were not entirely random, and were influenced by inherent phenotypic characteristics, management measures, and external environmental factors [2]. Therefore, to enhance the realism and diversity of the simulation environment, we implemented the following steps:At the beginning of each episode, a 10 × 2 m projection map was constructed, taking into account the real greenhouse plant distribution patterns. We inserted the curves representing the main stems of tomato plants (depicted as green lines) at regular intervals of 0.42 m into the map;Randomly, 0 to 5 target points (illustrated as blue dots) were selected on the main stem curves to represent the positions of tomato clusters. Notably, each point’s height was constrained within the range of 0.31 m to 0.79 m;In the map, a vehicle (represented by a gray rectangle) and its harvesting area (outlined by a red dashed line) were constructed. The vehicle was capable of moving left to right along the *X*-axis. If a target point fell within the harvesting area, the corresponding blue dot turned red, indicating the tomato had been harvested. Notably, each time the vehicle stopped, a black dot was generated on the *x*-axis to mark the parking location.

The simulation environment, as depicted in Figure 5, was constructed using the Python 2D plotting library Matplotlib.

##### State and Action Space

In this study, our primary objective was to train an intelligent agent capable of dynamically generating movement sequences for a dual-arm robot vehicle. This optimization aimed at ensuring that the robot halts precisely at locations that are most advantageous for performing tasks, such as harvesting, while also minimizing unnecessary parking to enhance overall efficiency. In the light of the above task, the state observed by the robot could be described as follows: (1) The robot location *x* in the *X*-axis (robot moving distance) where $x\in \left[0,10\right]$. (2) The number $n_{p}$ of times the robot has parked. (3) The number $n_{m}$ of missed target points. (4) The set $P=\left[P_{L},P_{R}\right]$ consisting of target points within the projection map, where each point is defined by the parameters [*x*, *h*, *state*]. Here, the element *state* is a Boolean in the state space, initially set to False, and switched to True when the point is within the harvest area. At time step *t*, the state observed from the environment is represented as a multi-dimensional concatenated vector:(6)$$s_{t}=\left[x,n_{p},n_{m},P\right]\in S,$$
Our goal was to plan the movement path of the robot vehicle within the projection map. Therefore, the actions output by the actor network correspond to the incremental displacement of the robot vehicle along the *X*-axis, as follows:(7)$$a_{t}=∆x\in A,$$
Simultaneously, we imposed constraints on the incremental values to prevent reversing and excessive movement distances. The range for these incremental values was accordingly set to $\left(0.1,\left.2.0\right]\right.$ m.

##### Reward Function

The critical aspect of this training was to optimize the robot’s operational efficiency by meticulously controlling its parking locations and the number of stops it makes. To guide this behavior, we designed three types of reward functions to encourage or penalize specific actions within the learning strategy, catering to different requirements: optimizing the parking locations ($r_{locat}$), reducing the number of parking events ($r_{park}$), and avoiding the omission of target points ($r_{miss}$).

The reward function for evaluating parking locations was defined in two ways: ensuring simultaneous operation of multiple arms as closely as possible, and the proximity of target points to the central axis within the harvesting area. Equation (8) describes the function employed for the parking location evaluation. Specifically, if any arm’s harvesting area lacked target points, the function assessed whether new target points could be added to this area without losing targets in other areas, and output an element Reval (Boolean type). If Reval was True, a significant negative value was assigned. Conversely, if all harvesting areas contained target points, the function calculated the average distance Dmean of these points from the central axis and encouraged proximity to the axis by rewarding the reciprocal of these distances. This procedure is summarized in Algorithm 2.
**Algorithm 2:** The parking location evaluation method.**Input**: robot locations *X*; the set P=PL,PR consisted of target points within the map; the set of target points within the harvesting area of the B1 arm PB1; the set of target points within the harvesting area of the B2 arm PB2; harvest area width *w*; the interspace between the B1 and B2 arms L.**Output**: Dmean; Reval
1:**if** PB1=empty and PB2=empty **then**2:Reval=True3:**else if** PB1≠empty and PB2=empty **then**4:Get the point pmin with minimum *x* in PB15:Get distance from pmin to the harvesting area’s left edge d=pmin−X−w/26:**if** X+w/2+L<PR.x<X+w/2+L+d **then**7:Reval=True8:**else**9:Reval=False10:**end if**11:**else if** PB1=empty and PB2≠empty **then**12:Get the point pmin with minimum *x* in PB213:Get the distance from pmin to the harvesting area’s left edge d=pmin−X+L−w/214:**if** X+w/2<PL.x<X+w/2+d **then**15:Reval=True16:**else**17:Reval=False18:**end if**19:**else if** PB1≠empty and PB2≠empty **then**20:Reval←False21:**for** each point in PB1
**do**22:DsumB1+=abspoint.x−X23:**for** each point in PB2
**do**24:DsumB2+=abspoint.x−X+L25:Dmean=DsumB1/num_PB1+DsumB2/num_PB226:**end if**27:**return** Reval; Dmean
(8)rlocat=−10                                if Reval=True0.4/Dmean+0.04     if Reval=False,
The reward function rpark was a decreasing nonlinear function of the number of stops, np as shown in Equation (9).
(9)rpark=−e0.1np,
For the reward function rmiss, we gave a larger penalty when target point was missed:(10)rmiss=−10nm       if nm≠00             if nm=0,
Finally, the total reward function for each step was as follows:(11)rtst,at=rlocat+rpark+rmiss.

### 2.5. Experimental Setup

We conducted both simulation and field experiments to evaluate the dual-arm robot vehicle control performance of the DDPG. The identification, simulation, and training activities detailed in this research were conducted using Python 3.7 and the PyTorch 1.11.0 framework on an Ubuntu 20.04 system. These processes were performed on a computer configured with an Intel i7-10700K CPU (Intel Corporation, Santa Clara, CA, USA), 32 GB RAM (Kingston Technology Company, Fountain Valley, CA, USA), and an Nvidia GeForce GTX 1080Ti GPU (Nvidia Corporation, Santa Clara, CA, USA).

#### 2.5.1. Simulation Experiments

In our simulation environment, we employed and compared three state-of-the-art DRL algorithms specifically tailored for continuous action spaces, for training purposes. These algorithms included Deep Deterministic Policy Gradient (DDPG) [28], Twin Delayed Deep Deterministic Policy Gradient (TD3) [31], and Soft Actor Critic (SAC) [32,33]. All these algorithms were configured with identical simulation environment settings and used the same dataset. We utilized the default initialization parameters and weights for each algorithm to guarantee their optimal performance. The key parameters of the simulation environment and the DDPG are detailed in Table 1.

Furthermore, we evaluated the performance of the DRL algorithm with reference to the metrics proposed in the reward function (Section 2.4.2). In the tables, results are presented for all 10 testing crop rows. The parameters that were compared are the number of stops $n_{p}$ required to achieve the goal, and the standard deviation $\sigma_{np}$ of the number of stops; the number of target points missed $n_{m}$, and the standard deviation $\sigma_{nm}$ of the missed number; the number of idle instances $n_{i}$ of the robot arm, and the standard deviation $\sigma_{ni}$ of the times $n_{ai}$; the average distance $D_{m}$ from the target points to the central axis in the working area, and the standard deviation $\sigma_{d}$ of the distance $D_{m}$. The smaller the distance $D_{mean}$, the better the parking location of the vehicle was.

#### 2.5.2. Field Experiments

In a real greenhouse environment, we also conducted comparative experiments to evaluate the effectiveness and efficiency of the vehicle movement planning system. We utilized a dual-arm robot as shown in Figure 2a, specifically designed for autonomous operations in greenhouse environments. It was capable of executing integrated tasks such as harvesting, collecting, and transporting while maneuvering between rows. Prior to the experiments, workers pruned side branches and leaves in the fruit maturation areas to avoid interference during the movement process.

For the real tests, we generated a tomato coordinate projection map using the method described in Section 2.3. The map was then input into three different algorithms to generate vehicle movement paths:Experiment 1—As illustrated in Figure 6, we used a traditional grid-based path planning algorithm [12,34] to generate vehicle movement paths. Based on the size of the arm’s harvesting area, the algorithm progressively constructed a global grid map and eliminated grids without target points. The intersection points of each grid’s central axis with the *X*-axis serve as the nodes for the generated movement paths;

2.Experiment 2—The process of the area division algorithm [13] is illustrated in Figure 7. Initially, the closest fruit coordinate to the origin was chosen as area 1’s center (p0), with its effective picking area marked, including points p1 and p2 (Figure 7a). Next, areas centered on p1 and p2 were compared. p2, with more fruit points, became the new center (Figure 7b–d). This process was repeated for subsequent areas, starting from the closest point to the previous center (Figure 7e–h). The intersection points of each area’s central axis with the *X*-axis serve as the nodes for the generated movement paths;

3.Experiment 3—The final experiment utilized the vehicle movement planning method grounded in DRL, as outlined in this paper. It aimed to dynamically generate the action sequence of the dual-arm robot vehicle, encompassing its incremental displacement and the number of stops.

## 3. Results and Discussions

### 3.1. Simulation Experiments

Operating within the same simulation environment, Figure 8 illustrates the learning curves of six distinct models—DDPG (our), SAC, and TD3—during their learning phase. The results reveal that, overall, DDPG (the red curves) outperformed all other evaluated algorithms, demonstrating superior convergence performance in the specified task of vehicle movement planning. Specifically, DDPG and SAC reached a stable state at around 1400 episodes, achieving a learning rate that was 80% faster than the approximately 7000 episodes required by TD3. Furthermore, DDPG improved by approximately 19.82% over SAC and 33.66% over TD3 in terms of the average return when reaching stability.

Table 2 presents the simulation results of the trained policy model based on a test set of 10 crop rows. Our DDPG decision model demonstrated outstanding performance in intermittent stop-and-move motion planning. In a 10 m crop row, it averaged 27.4 ± 0.5 parking instances, missed only 0.4 ± 0.7 target points on average, and experienced 3.3 ± 0.8 idle arm instances. Importantly, the average distance of target points from the central axis within work areas was just 0.096 ± 0.005 m, showcasing the model’s ability to effectively plan optimal parking positions. However, there was a drawback: compared to the TD3, which had the lowest parking frequency, our model’s parking frequency increased by 17.3%. It was noticeable that the lower parking frequency comes at the cost of a reduced harvest yield. Subsequently, the vehicle needed to return to the missed locations, which ultimately hindered the overall operational efficiency.

### 3.2. Field Experiments

In a real greenhouse setting, we conducted vehicle movement experiments using an integrated cherry tomato-harvesting robot system capable of recognition and motion decision-making. The movement process of the robot was as follows: Initially, as the robot moved, it employed the visual system (as detailed in Section 2.3) to track tomato clusters within a specific area of the image. Once a tomato cluster met the positioning criteria (Algorithm 1), the robot then acquired the 3D coordinates of the target tomato relative to the base coordinate system of the arm and the moving distance. After moving 12 m, the robot stopped, and converted the collected coordinate points and the distance into a projection map. Subsequently, this projection map of the first 10 m was input into the above three experimental algorithms described in Section 2.5.2. Finally, the vehicle moved in reverse along the planned path, returning to the starting point.

Within the 10 m crop row, the projection map recorded a total of 106 target points. Compared to the actual count of 110 target tomato clusters, this resulted in a statistical accuracy rate of 96.4%. The results of the field experiments for automatic vehicle movement are presented in Table 3 and Figure 9. We analyzed the movement paths of the dual-arm robot vehicle under different experimental schemes by comparing the vehicle parking times, the numbers of missed targets, the numbers of idle times of the arms, the parking locations and the process speeds in the same environment.

In Experiment 1, we created a grid map for planning the movement path for the dual-arm robot vehicle. The results show that to return to the starting position without missing any targets, the vehicle needed to stop 43 times, but the path generation only took 3.7 ms. Notably, during each stop, only one arm was operational. This was due to the distance between the two arms not being an integer multiple of the arm’s working area. When one arm aligned with the central axis of the working area, it inevitably led to the misalignment of the other arm with the working area (Figure 9a). Additionally, the grid map was unable to effectively distribute targets within each work area, resulting in a maximum $D_{m}$. In Experiment 2, we implemented an area division algorithm as described in Section 2.5.2. This algorithm allowed for the more rational planning of parking locations, achieving the smallest average distance from target points to the central axis in the work area ($D_{m}$ minimum). However, due to the need for multiple cycles and queries, the time required to generate the path increased to 14.1 ms. Additionally, there was still an issue with comprehensively planning the positions of the working areas on both sides. Out of 36 parking instances, in 34, only one arm was operational.

Finally, in Experiment 3, we integrated the DDPG decision model and all methods proposed in this paper for the final experiment on the robot vehicle, as shown in Figure 9c. The results indicate that returning to the starting position without missing any targets required only 23 stops. This setup allowed for the comprehensive consideration of work areas on both sides when planning parking locations, reducing the number of idle instances of the arms to four. The average distance from target points in the work area to the central axis was 0.093 m, and the processing time was only 3.2 ms longer than the fastest grid map method. Overall, our model achieved infrequency movement planning for the vehicle, logically planned optimal parking locations, and enhanced the overall operational efficiency.

### 3.3. Discussion and Future Work

The present study has demonstrated the efficacy of real-time projections of tomato clusters and a vehicle movement planning algorithm based on DRL in a greenhouse setting. In the context of vehicle parking node-planning tasks, the DDPG policy model proved to be effective in determining the parking locations and frequencies for a dual-arm robot amidst crop rows. In this study, we only tried to build a typical DRL model, and future explorations could include integrating Long Short-Term Memory (LSTM) layers into the actor or critic networks [15], or establishing a multi-agent reinforcement learning framework [35].

Furthermore, our method generated a 10 m-long projection map before deciding on vehicle movements, which may not be useful for real-time motion planning. Future research will aim to ascertain the ideal size of the projection map that would enable the decision model to efficiently plan vehicle movements both in real time and effectively.

## 4. Conclusions

To address the efficiency demands of dual-arm harvesting robots, we proposed a novel parking node planning method grounded in Deep Deterministic Policy Gradient (DDPG). The policy model, predicated on the spatial distributions of fruit, generated a sequence of strategic parking locations. This approach sought to strike a balance between minimizing parking instances and ensuring an even coverage of randomly distributed fruits on both sides, thereby facilitating the robot’s efficient operation.

The proposed method integrated considerations of frequent stops, missed targets, and uneven task allocation, outperforming both SAC and TD3 in terms of convergence performance. Furthermore, the field experiment results demonstrate that the DDPG policy model with parking constraints significantly improved efficiency: the model reduced the parking frequency of the vehicle by 46.5% and 36.1%, and decreased the arms’ idle instances by 42.9% and 33.9%, respectively, in comparison to the grid-based and area division-based planning algorithms. Importantly, these improvements were realized without any instances of missed targets. Additionally, the average time required to generate parking node sequences was 6.9 ms.

In conclusion, the DRL method demonstrated its robustness and efficiency in planning parking nodes for dual-arm harvesting robots. This has provided novel perspectives on the development of intelligent agricultural robotics. Future works will incorporate dual-arm coordination and sequenced harvest stops into our reinforcement learning model. Additionally, we aim to design a more advanced version of this model to further improve overall operational efficiency.

Our method proved robust and efficient for the intermittent stop–move planning of dual-arm harvesting robots, providing innovative insights into intelligent agricultural robotics. In future work, we will extend our reinforcement learning model to include dual-arm coordination and sequenced harvest stops. We will also enhance the generalization of our model to multiple complex navigation scenarios, as well as other types of produce with varying target densities.

## Figures and Tables

**Figure 1 biomimetics-09-00105-f001:**
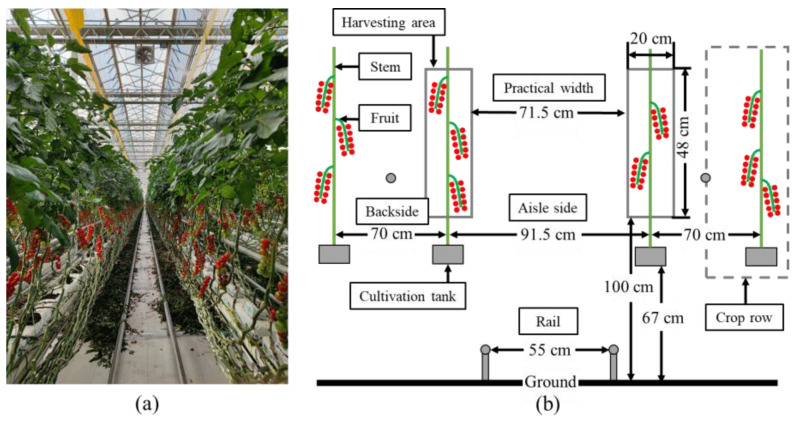
Overview of the experimental environment. (**a**) Greenhouse environment. (**b**) Schematic side view of the crop.

**Figure 2 biomimetics-09-00105-f002:**
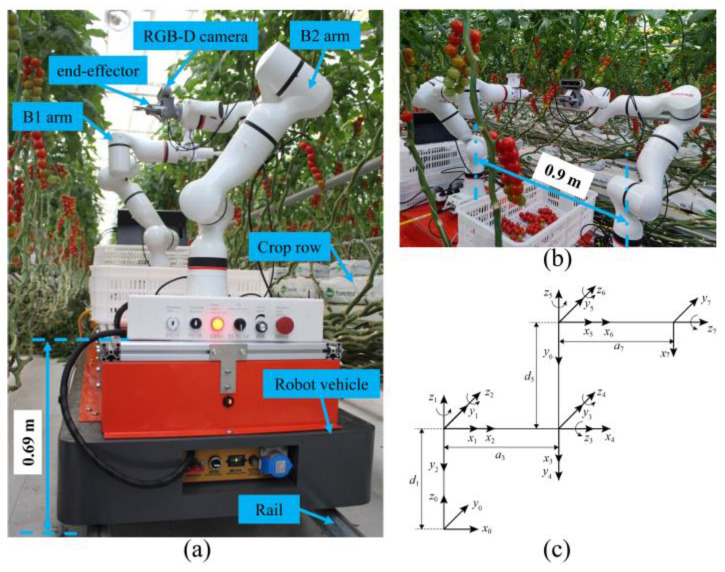
Overview of the dual-arm tomato-harvesting robot. (**a**) Prototype. (**b**) Side view of the robot. (**c**) Link coordination system of the arm.

**Figure 3 biomimetics-09-00105-f003:**
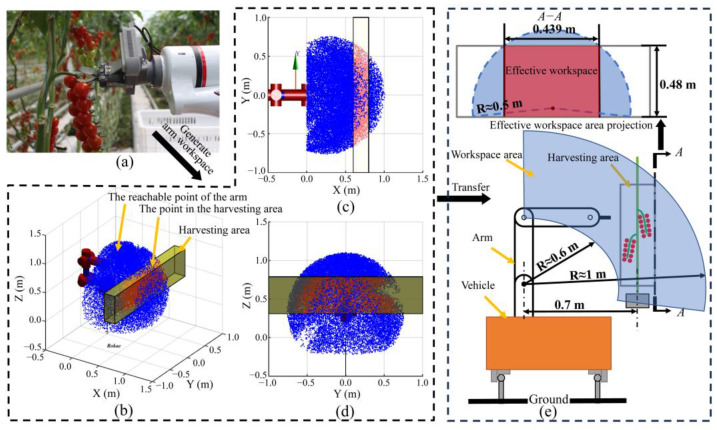
Effective workspace area of the harvesting arm. (**a**) Example of the robot in a harvesting posture. (**b**) Workspace of the arm with a constant picking pose. (**c**) Top view (X−Y Plane) of the workspace. (**d**) Side view (Y−Z Plane) of the workspace. (**e**) Schematic of the operating area in a greenhouse environment. (*A*–*A*) Projection of the effective workspace area.

**Figure 4 biomimetics-09-00105-f004:**
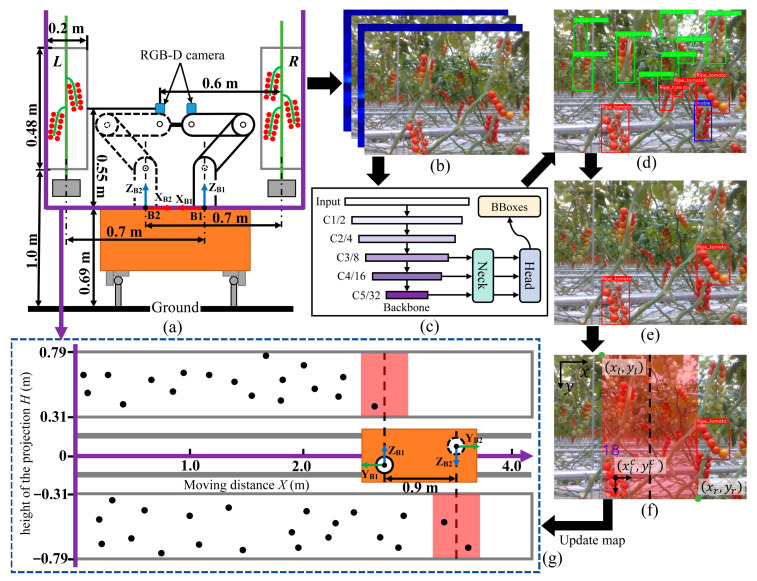
The projection process of ripe tomato clusters on the two-dimensional crop row map. (**a**) Side view of the dual-arm robot between the crop rows. (**b**,**c**) Original color and aligned depth images of the crop rows captured by cameras. (**d**) Detection results using YOLOv5 model: green boxes indicate unripe tomato clusters, red for ripe tomato clusters, and blue for ripe tomato clusters unfit for harvesting. (**e**) Identified harvesting targets after the depth threshold processing. (**f**) Tracking area for tomato cluster localization. (**g**) Map coordinate system.

**Figure 5 biomimetics-09-00105-f005:**
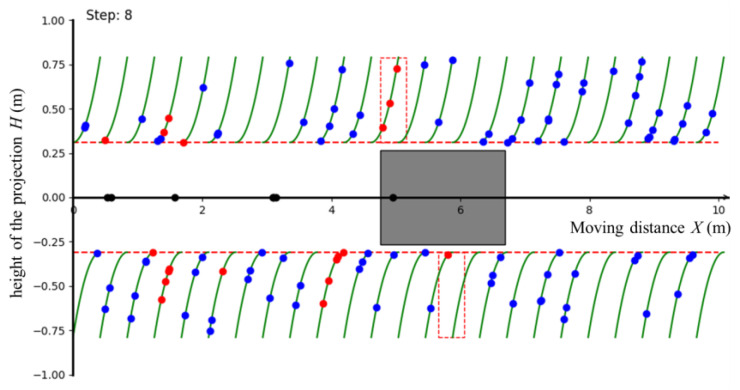
The interaction environment. The red rectangle represents the effective working area projection of the arms, and the gray rectangle represents the vehicle. The red dot indicates that the target point has appeared in the robot arm operating area. Red dots indicate harvested targets, blue dots indicate missed or unharvested targets. The green line represents the main stem, and all target points only appear on the main stem. The red line represents the boundary line.

**Figure 6 biomimetics-09-00105-f006:**
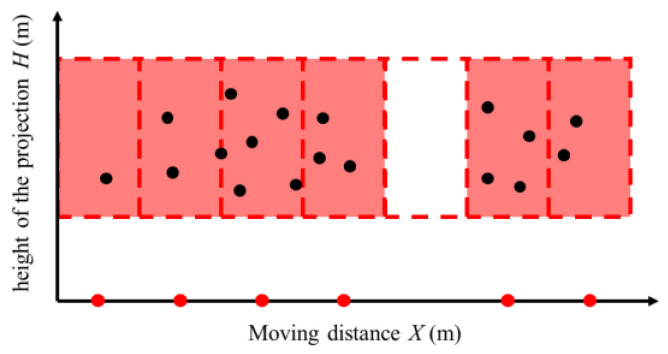
Grid-based path planning algorithm. The red rectangle represents the harvesting area, the dotted line represents the divided grid map, and the red dot represents the movement nodes.

**Figure 7 biomimetics-09-00105-f007:**
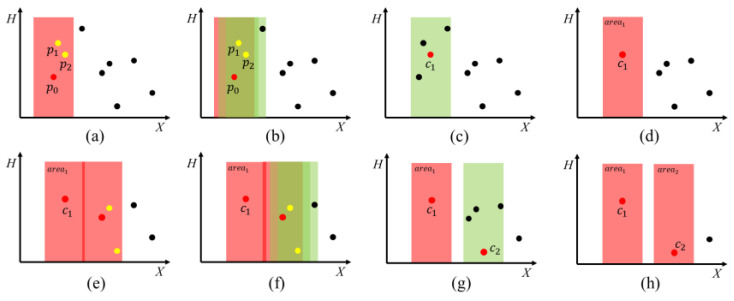
The process of the area division algorithm. (**a**) Determine the initial harvesting area; (**b**) Generate candidate areas; (**c**) Select the candidate area containing the most target points (**d**) Determine the first harvesting area at the current location. (**e**–**h**) Repeat the operation of (**a**,**b**). The red rectangle represents the harvesting area, and the green rectangle represents the candidate area. The red dot indicates that the harvesting area is centered on this point. The yellow dot indicates that the candidate area is centered on this point. Black dots indicate target points to be harvested.

**Figure 8 biomimetics-09-00105-f008:**
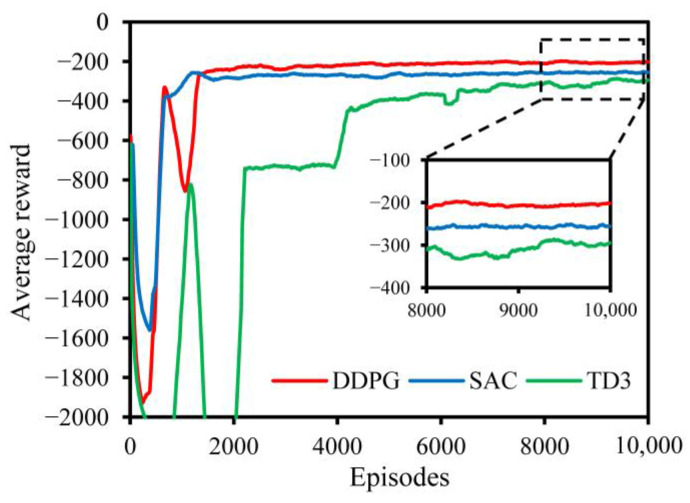
The learning curves of the training process for the vehicle movement planning task with different DLR algorithms.

**Figure 9 biomimetics-09-00105-f009:**
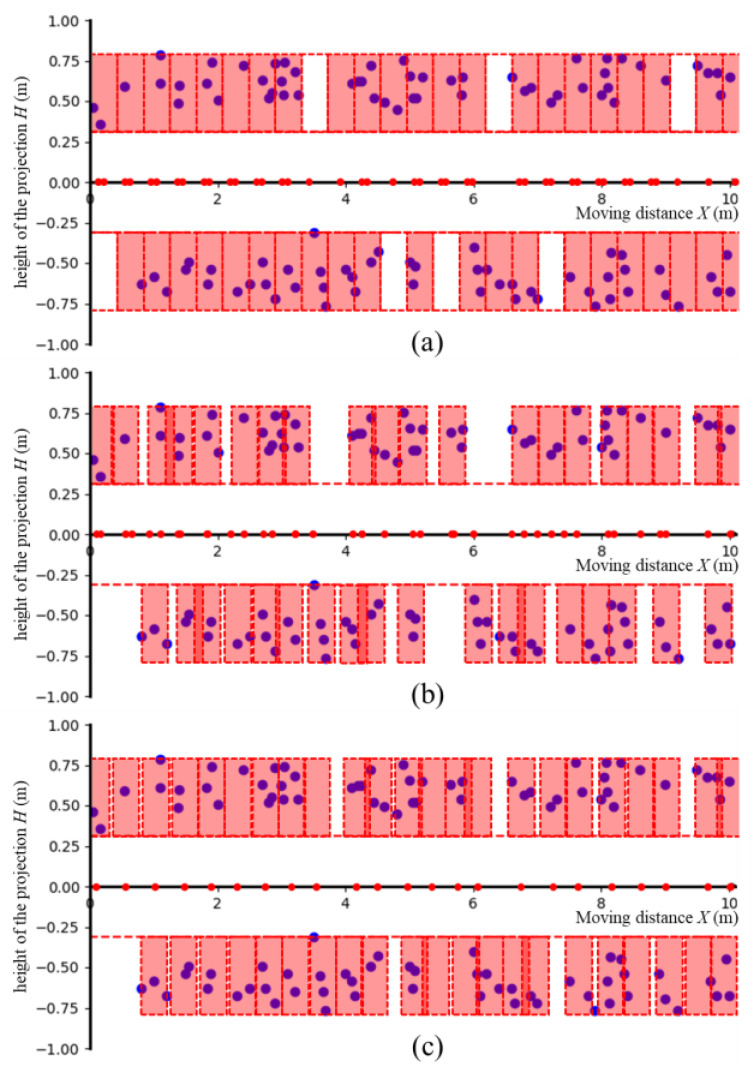
The vehicle motion planning paths generated by the three algorithms. (**a**) Grid-based path-planning algorithm. (**b**) Area division algorithm. (**c**) DDPG-based path-planning algorithm. The red dot indicates the parking location of the vehicle in the crop row. The red rectangle represents the divided working area. The purple dot represents the projected coordinates of the tomato clusters that are ready to be harvested.

**Table 1 biomimetics-09-00105-t001:** The key parameters of the simulation environment and the DDPG.

Object	Parameters	Value
Simulation environment	Robot vehicle size	1.94×0.53 m
	Vehicle driving distance	10 m
	Arm harvest area	0.4 × 0.48 m
	Distance between the two arms	0.9 m
	Number of target points	100~120
	Left crop row tomato cluster projection area	x∈[0,10] m, h∈[0.31,0.79] m
	Right crop row tomato cluster projection area	x∈[0,10] m, h∈[−0.79,−0.31] m
DDPG	Number of episodes	10^5^
	Max episode step	50
	Mini-batch Size	128
	Discount factor	0.99
	Decay coefficient	0.0001
	Soft update factor	0.001
	Replay buffer size	10^6^
	Learning rate of actor network	0.0001
	Learning rate of critic network	0.001
	Optimizer for SGD	Adam [34]

**Table 2 biomimetics-09-00105-t002:** The simulation results of the three distinct models on the test set.

Algorithm	$\mathrm{avg}.\;n_{p}\pm \sigma_{np}$ ^1^	$\mathrm{avg}.\;n_{m}\pm \sigma_{nm}$ ^1^	$\mathrm{avg}.\;n_{i}\pm \sigma_{ni}$ ^1^	$\mathrm{avg}.\;D_{m}\pm \sigma_{D}$/m ^1^
DDPG	27.4 ± 0.5	0.4 ± 0.7	3.3 ± 0.8	0.096 ± 0.005
SAC	25.5 ± 0.5	8.2 ± 4.1	4.9 ± 1.4	0.100 ± 0.007
TD3	22.7 ± 0.9	17.2 ± 3.5	4.2 ± 1.9	0.103 ± 0.006

^1^ The definitions of parameters are described in detail in Section 2.5.1.

**Table 3 biomimetics-09-00105-t003:** The results of the three algorithms on the test crop row.

Algorithm	$n_{p}$ ^1^	$n_{m}$ ^1^	$n_{i}$ ^1^	$D_{m}$/m ^1^	Process Speed/ms ^1^
Grid map	43	0	43	0.103	3.7
Area division	36	0	34	0.080	14.1
DDPG	23	0	4	0.093	6.9

^1^ The definitions of parameters are described in detail in Section 2.5.1. ^2^ The speed represents the average time required to plan the entire path.

## Data Availability

The data presented in this study are available on request from the corresponding author.

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
