# Peer review of "Intermittent Stop-Move Motion Planning for Dual-Arm Tomato Harvesting Robot in Greenhouse Based on Deep Reinforcement Learning"

_biomimetics, 2024, doi:10.3390/biomimetics9020105_

Round 1

Reviewer 1 Report

Comments and Suggestions for Authors

The paper compares DDPG, TD3, SAC for intermittent motion planning in an agricultural application (dual arm harvesting robot). The paper shows the attractive performance of DDPG, enhancing vehicle and arm idleness, and the efficiency in path planning.

[1] On literature, the related works on motion planning using have been largely ignored. The topic of of dual-arm mobile robot motion planning has received considerable attention in the community in recent years. It would be desirable to (conceptually) compare/describe the taxonomy of the existing state of the art approaches (in a table or picture) and highlight the key contributions of the paper.

[2] It is unclear whether the suggested approach is generalizable over multiple complex navigation scenarios. Only a number of preliminary experiments have been presented using a few navigation scenarios. It would be desirable to show the performance over several navigation cases, independent runs, and several origin-destination pairs. Furthermore, a comparison with state-of-the-art types of combinatorial motion planning frameworks would be most desirable.

The paper has the potential for simple and efficient dual-arm motion planning for static/dynamic environments, yet evaluations and comparisons are still a weak point in the manuscript. Without rigorous evaluations over various challenging navigation scenarios, and comparisons to other combinatorial approaches, it is hard to conclude that the observations proposed in the paper contribute to the field substantially or are relevant for practical use.

Comments on the Quality of English Language

Minor edits on the overall grammar is desirable

Author Response

Dear Reviewer:

Thank you very much for your comments concerning our manuscript. Those comments are all valuable and very helpful for revising and improving our paper, as well as the important guiding significance to our researches. We have tried our best to revise our manuscript according to your comments. Any revised portion made to the manuscript are highlighted in red. Please see the attachment for the main revisions in the paper and the responses to the reviewers' comments.

Thank you very much, and look forward to your reply! I wish 2024 will bring you prosperity and good fortune.

Best regards,

Qingchun Feng

Reviewer 2 Report

Comments and Suggestions for Authors

The manuscript presents a method for intermittent stop-move motion planning. It is applied to the control of a harvesting robot. Deep reinforcement learning is used by the authors.

The approach is interesting and I have found some merit in the work. There are some issues that the authors should address to clearly explain their contributions.

- At the end of section 1, the authors should clearly and concisely state which are the contributions, differences and advantages of their proposal with respect to other recent related works in the state of the art.

- In section 2.3, it would be nice if the authors describe which is the performance of their proposal with respect to the detection of the tomato clusters depending on the speed of the camera, the lighting conditions of the environment and the ripeness of the fruits. Have the authors tested their approach in presence of blur produced by the movement of the camera?. Apart from the precision, it would be useful to know the recall of the proposal.

- It would be nice if the authors include their proposal (simulator and algorithms) in a public repository in such a way that further research on this topic can have a reference work to compare with.

- In the experimental section, I would like to see the influence of the density of tomato clusters (or the total number in a given area). This way, the reader could understand if the proposal can be used directly with other kinds of vegetables or fruits with substantially higher or lower density of targets.

- Are the results very sensitive to the key parameters shown in the second part of table 1?

Author Response

(The authors gave the same response as above.)

Round 2

Reviewer 1 Report

Comments and Suggestions for Authors

The paper has been revised by including further descriptions of literature and additional revision of English.

The revisions, although desirable, lack rigorous performance evaluations over several navigation cases and independent runs. It would be preferable to evaluate the algorithms over independent runs through arbitrary navigation/collection cases (the few examples in the paper give an idea that the results were selected by a "cherry-picking" approach).

As stated in a previous revision, the paper has the potential for simple and efficient dual-arm motion planning, yet evaluations and comparisons are still a weak point in the current version. It is highly recommended to perform rigorous evaluations over various challenging navigation scenarios to conclude and convince readers that the DRL approach contributes to the field substantially or is relevant for practical use in agriculture applications.

Author Response

Dear Reviewer:

Thank you very much for providing the opportunity to revise once again. Those comments are all valuable and very helpful for revising and improving our paper, as well as the important guiding significance to our researches. We have tried our best to revise our manuscript according to your comments. Any revised portion made to the manuscript are highlighted in red. Please see the attachment for the main revisions in the paper and the responses to the reviewers' comments.

Thank you very much, and look forward to your reply! I wish 2024 will bring you prosperity and good fortune.

Reviewer 2 Report

Comments and Suggestions for Authors

The authors have considered just partially the comments I made in my previous review. In my opinion, there is still room for improvement, and the authors should put more effort in presenting the main features and superiority of their proposal.

- In my previous review I stated "In section 2.3, it would be nice if the authors describe which is the performance of their proposal with respect to the detection of the tomato clusters depending on the speed of the camera, the lighting conditions of the environment and the ripeness of the fruits. Have the authors tested their approach in presence of blur produced by the movement of the camera?. Apart from the precision, it would be useful to know the recall of the proposal". The answer of the authors is just a sentence to say that the cameras travel at 2m/s and that the precision is 94.4% and the F1-score 95.7%. Therefore, the authors do not study the influence of the speed of the camera, or the ripeness of the fruits or the lighting conditions. Some additional experiments should be performed to clearly prove the robustness of this part of the framework.

- In my previous review, I recommended the authors to include their proposal (simulator and algorithms) in a public repository. They write that they will upload them to GitHub, but no further information or link is included.

- I keep on thinking that studying the influence of the density of fruits would be interesting to the reader. The authors have taken no action on this suggestion.

- Finally, I also asked if the results were sensitive to the key parameters (second part of table 1). The authors should experimentally show it. In my opinion, it is important for the reader to know if the approach must be carefully tuned for every new scenario, or if can be extended to other environments more directly.

Author Response

(The authors gave the same response as above.)
